# WHICH SPARSE CODE? IDENTIFIABILITY FAILURES IN SAE INFERENCE

## ABSTRACT

Sparse autoencoders (SAEs) are widely used in mechanistic interpretability, but it is unclear whether the encoder's sparse code is uniquely determined. We compare SAE encoders against classical sparse coding algorithms (OMP, IHT) using frozen dictionaries. We find that alternative methods select substantially different features (Jaccard $\sim 0.43$) while producing linearly equivalent codes ($R^2 > 0.88$). This dissociation between linear and support identifiability holds across layers and SAE configurations. Our results suggest SAE features represent one valid decomposition among alternatives, with implications for interpretability claims built on specific features.

## 1 INTRODUCTION

Sparse Autoencoders (SAEs) have become a key tool for decomposing neural network activations into interpretable features (Cunningham et al., 2023; Bricken et al., 2023). SAE-based interpretability practice implicitly reduces encoder features as canonical as specific features are cataloged, named, and used for causal interventions. But whether the encoder's particular solution is preferred among valid sparse codes is rarely examined. Recent findings suggest that this assumption deserves scrutiny. Heap et al. (2025) demonstrated that SAEs trained on *randomly initialized* transformers produce features with interpretability scores comparable to those trained on actual models. Paulo & Belrose (2025) showed that SAEs trained on the same data with different random seeds share only $\sim 30\%$ of features. These results point to a broader concern: we cannot reliably predict when interpretability methods will yield stable, unique solutions. The problem isn't just that different training runs produce different SAEs—it's that we lack guarantees about when any particular solution is privileged.

We ask a complementary, inference-time question: *For a single, fixed SAE, does the encoder's sparse code represent the unique valid solution, or do alternative codes exist that reconstruct equally well?* If alternatives exist, what is the relationship between them? We test this by comparing the encoder against alternative sparse coding algorithms: Orthogonal Matching Pursuit (OMP), Iterative Hard Thresholding (IHT), and gradient descent with sparsity penalties – operating on the same frozen SAE dictionary. We measure not just reconstruction quality and feature overlap, but also linear identifiability, stability under noise perturbations, and the uniqueness of the decoder itself. Our findings reveal that SAE representations are non-unique at multiple levels, with implications for how we interpret SAE-based analyses.

### 1.1 BACKGROUND

**Sparse Autoencoders.** Sparse autoencoders (SAEs) decompose neural activations into sparse, interpretable components. Given an activation $\mathbf{x} \in \mathbb{R}^d$ (e.g., the residual stream at some layer), an SAE reconstructs it through a sparse code $\mathbf{z} \in \mathbb{R}^m$:

$$\mathbf{z} = \mathrm{ReLU}\left(W_{\mathrm{enc}}\mathbf{x} + \mathbf{b}_{\mathrm{enc}}\right), \qquad \hat{\mathbf{x}} = W_{\mathrm{dec}}\mathbf{z} + \mathbf{b}_{\mathrm{dec}}. \tag{1}$$

ReLU enforces non-negativity and sparsity, so most entries of $\mathbf{z}$ are zero. The decoder reconstructs activations as a linear combination of dictionary columns $\mathbf{d}_i$ (the columns of $W_{\mathrm{dec}}$): if $\mathbf{z}$ is supported on $\{i_1, \ldots, i_k\}$, then $\hat{\mathbf{x}} \approx \sum_j z_{i_j} \mathbf{d}_{i_j}$. The dictionary is typically overcomplete ($m \gg d$, often 8–32×), and training minimizes reconstruction loss plus a sparsity penalty on $\mathbf{z}$.

**Linear Identifiability.** Roeder et al. (2021) prove that learned representations are *linearly identifiable*: models trained on the same data learn representations related by an invertible linear transformation. Thus, the information captured is identical and only the coordinate system differs. They measure this via Canonical Correlation Analysis (CCA), which finds linear projections that maximize correlation between representation sets. We adapt this framework to sparse codes: given codes produced by different algorithms on the same dictionary, we measure $R^2$ (can one code predict the other via linear regression?) and CCA (do they span the same subspace?). We also report Cosine Similarity, and Jaccard similarity, the fraction of shared active features. The first two capture linear equivalence, while the last captures whether methods select the same dictionary elements.

## 1.2 LEARNING SPARSE CODES

**Setup** We collect activations by running Gemma 2 on $\sim$600 factual knowledge prompts. For the SAEs, we use GemmaScope (Lieberum et al., 2024), which provides JumpReLU SAEs for Gemma 2 2B residual-stream activations, with widths $m \in \{16k, 65k\}$ across layers $\ell \in \{5, 12, 20\}$. For an activation $\mathbf{x} \in \mathbb{R}^d$ (e.g., $d$=2304 for Gemma 2 2B), the encoder produces a sparse code $\mathbf{z} \in \mathbb{R}^m$ (e.g., $m$=16,384):

$$z_i = \begin{cases} \text{ReLU}(\mathbf{w}_i^\top \mathbf{x} + b_i) & \text{if } \mathbf{w}_i^\top \mathbf{x} + b_i > \theta_i, \\ 0 & \text{otherwise}, \end{cases} \quad (2)$$

where $\mathbf{w}_i$ is the $i$-th encoder row, $b_i$ is the corresponding bias, and $\theta_i$ is a learned threshold. We denote the number of active features as $k = \|\mathbf{z}\|_0$. Our dataset consists of factual completion prompts spanning knowledge retrieval, arithmetic, and linguistic tasks, chosen for clear ground truth in causal experiments.

**Alternative sparse codes.** For a fixed activation $\mathbf{x}$ and frozen decoder $W_{\text{dec}} \in \mathbb{R}^{m \times d}$ (whose rows are learned dictionary directions), we compare the encoder's code $\mathbf{z}_{\text{enc}}$ against classical sparse coding algorithms with known theoretical properties (Appendix A). **All methods are constrained to select exactly $k$ features, matching the encoder's sparsity.**

*Orthogonal Matching Pursuit (OMP)* (Pati et al., 1993) is a greedy method with recovery guarantees under restricted isometry conditions (Tropp, 2004).Starting from residual $\mathbf{r} = \mathbf{x}$, it iteratively (i) selects the dictionary row most correlated with $\mathbf{r}$, (ii) refits coefficients by least squares over selected rows, and (iii) updates $\mathbf{r}$ to the remaining reconstruction error. Unlike the encoder's single-shot prediction, OMP reasons sequentially. We additionally test OMP-Exclude-Top-$N$ (Appendix B), which forbids the encoder's top features, directly probing alternative feature sets with comparable reconstruction.

*Iterative Hard Thresholding (IHT)* alternates gradient steps on reconstruction error with hard thresholding (Blumensath & Davies, 2008): each iteration takes a gradient step on $\|\mathbf{x} - W_{\text{dec}}^\top \mathbf{z}\|^2$, then zeros all but the $k$ largest entries of $\mathbf{z}$. We test initialization from $\mathbf{z}_{\text{enc}}$ (probing whether the encoder found a local optimum) and from random $k$-sparse vectors (probing distinct basins).

*Gradient descent with $\ell_1$ penalty* optimizes $\min_{\mathbf{z}} \|\mathbf{x} - W_{\text{dec}}^\top \mathbf{z}\|^2 + \lambda \|\mathbf{z}\|_1$, where $\lambda$ controls sparsity. We adapt $\lambda$ during optimization: every 50 steps, if $\|\mathbf{z}\|_0 > 1.2k$ we increase $\lambda$, and if $\|\mathbf{z}\|_0 < 0.8k$ we decrease it. After convergence, we hard-threshold to exactly $k$ active features. Multiple random initializations probe distinct local minima.

## 1.3 EXPERIMENTAL QUESTIONS

**How do codes relate?** Given encoder code $\mathbf{z}_{\text{enc}}$ and alternative code $\mathbf{z}_{\text{alt}}$, we measure both reconstruction quality and code similarity: (i) *Jaccard similarity* over supports $S_{\text{enc}} = \{i : z_{\text{enc},i} > 0\}$ and $S_{\text{alt}}$, i.e. $|S_{\text{enc}} \cap S_{\text{alt}}|/|S_{\text{enc}} \cup S_{\text{alt}}|$; (ii) *linear $R^2$*, by regressing $\mathbf{z}_{\text{enc}}$ on $\mathbf{z}_{\text{alt}}$ and vice versa and averaging; high $R^2$ indicates linear predictability even if supports differ; and (iii) *CCA*, which finds projections maximizing correlation between code sets; CCA $\approx 1$ indicates a shared linear subspace (Roeder et al., 2021). Since $n \approx 500 \ll m$ (e.g., $m$=16,384), we first project onto 50 principal components ($\sim 85\%$ variance retained).

**Are the codes stable under Gaussian noise?** To test robustness to input perturbations, we add Gaussian noise $\mathbf{x}_{\text{noisy}} = \mathbf{x} + \sigma\epsilon$, where $\epsilon \sim \mathcal{N}(0, I)$, and report Jaccard similarity between $\mathbf{z}(\mathbf{x})$ and $\mathbf{z}(\mathbf{x}_{\text{noisy}})$ across $\sigma \in [0.01, 1.0]$. If codes are stable (high Jaccard under noise), observed non-

uniqueness across methods reflects multiple valid solutions rather than optimization instability; if unstable, the encoder may select arbitrary solutions among many nearby alternatives.

**Does non-uniqueness extend to decoders?** The above evaluates uniqueness of codes for a fixed decoder. We also ask whether the decoder itself is uniquely determined by the codes. Given activations $X \in \mathbb{R}^{n \times d}$ and codes $Z \in \mathbb{R}^{n \times m}$ from any method, we fit an optimal decoder by ridge regression:

$$D^* = \arg\min_D \|X - ZD\|^2 + \lambda\|D\|^2, \tag{3}$$

where $D \in \mathbb{R}^{m \times d}$ maps codes to reconstructions and $\lambda$ provides numerical stability. We measure cosine similarity between learned decoder rows $D_i^*$ and original SAE decoder rows $(W_{\text{dec}})_i$. If the trained decoder is uniquely determined by the codes, similarity should be high ($\sim 1$); low similarity suggests multiple valid decoders exist, extending non-uniqueness to the learned dictionary directions themselves.

## 2 RESULTS

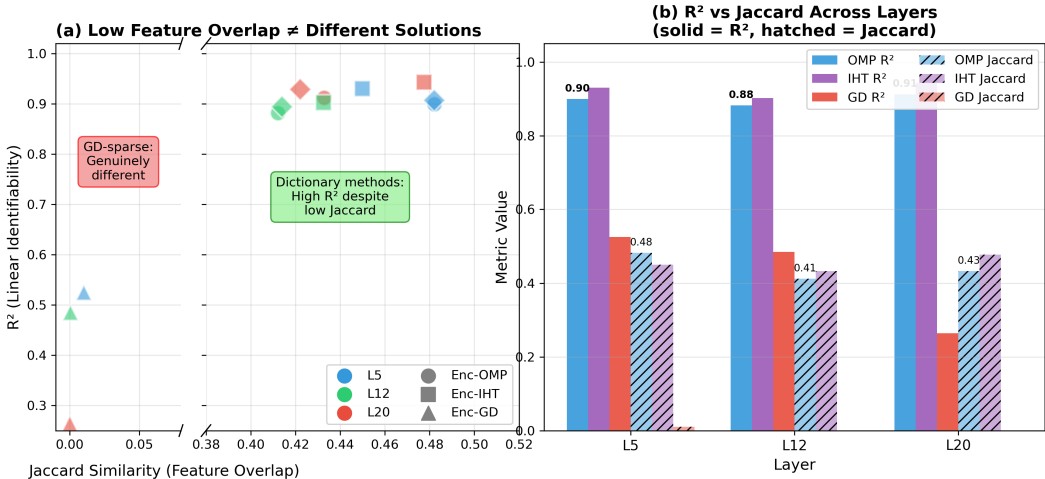

Figure 1: Summary of results.

Table 1 and Figure 1 summarize our findings across GemmaScope SAEs for Gemma 2 2B at layers 5, 12, and 20.

Table 1: Sparse coding results on GemmaScope SAEs (Gemma 2 2B). OMP refers to standard OMP without feature exclusion.

| SAE | Layer | OMP Recon | OMP Jacc | OMP $R^2$ | IHT Recon | IHT Jacc | IHT $R^2$ | GD Recon | GD Jacc | GD $R^2$ |
|-----|-------|-------|------|-------|-------|------|-------|-------|------|-------|
| 2B-16k | L5 | 0.93 | 0.48 | 0.90 | 1.02 | 0.45 | 0.93 | 2.52 | 0.01 | 0.52 |
| 2B-16k | L12 | 0.93 | 0.41 | 0.88 | 0.99 | 0.43 | 0.90 | 1.90 | 0.00 | 0.48 |
| 2B-16k | L20 | 0.93 | 0.43 | 0.91 | 0.99 | 0.48 | 0.94 | 2.06 | 0.00 | 0.26 |

**Linear identifiability holds, support identifiability fails.** OMP and IHT produce codes that are linearly equivalent to the encoder's ($R^2 > 0.88$, CCA $> 0.99$) despite selecting different features (Jaccard $\sim 0.43$). The codes span the same subspace but disagree on which dictionary elements to use. Notably, OMP achieves 7% better reconstruction than the encoder while using a largely disjoint feature set—the encoder's solution is not uniquely optimal.(Appendix B).

**GD-sparse finds qualitatively different (and worse) solutions**

Unlike OMP and IHT, gradient descent with an $\ell_1$ penalty converges to codes that are neither linearly equivalent to the encoder ($R^2 = 0.26$–$0.52$) nor competitive in reconstruction quality ($1.9$–$2.5\times$ worse). The near-zero Jaccard similarity ($\sim 0.00$) indicates these codes share essentially no features with the encoder. We attribute this to the mismatch between soft and hard sparsity constraints: features optimized jointly under $\ell_1$ penalties may not remain optimal when hard-thresholded to exactly $k$ active features. Though this instability may in itself informative as it suggests the sparse coding landscape contains distinct basins. See Appendix A for more details.

**Non-uniqueness is not instability.** Despite the existence of alternatives, encoder codes are highly stable: at noise level $\sigma = 0.1$, perturbed inputs yield codes with Jaccard $> 0.97$ relative to clean inputs (Appendix E). The encoder consistently finds its particular solution; other methods consistently find different ones.

**Decoder non-uniqueness.** When we learn optimal decoders for each method's codes via least squares, similarity to the original SAE decoder is low ($\sim 0.13$–$0.15$ cosine similarity). Non-uniqueness extends beyond feature selection to the dictionary directions themselves (Appendix F).

**Causal relevance.** Preliminary ablation experiments on layer 20 suggest these differences have behavioral consequences: ablating OMP's top-5 features disrupts model predictions at least as often as ablating the encoder's (42.9% vs. 40.5% breaking rate), despite only 43% feature overlap (Appendix G).

## 2.1 DISCUSSION

**Implications for interpretability.** Our results reveal a tension at the heart of SAE-based interpretability. Linear identifiability holds: different methods find codes containing equivalent information ($R^2 > 0.88$). But support identifiability fails: those codes select different features (Jaccard $\sim 0.43$). For analyses that depend only on the subspace spanned by active features—such as probing or linear classification—this non-uniqueness may be benign. But interpretability practice typically makes claims about *specific* features: dashboards like Neuronpedia (Lin, 2023) catalogue individual features, researchers name them (e.g., "the Golden Gate Bridge feature"), and interventions target particular dictionary directions. Our findings suggest such claims require qualification—alternative valid feature sets exist.

**Multiple basins, not just coordinate ambiguity.** The GD-sparse results ($R^2 \sim 0.26$–$0.52$) indicate that the sparse coding landscape contains genuinely distinct solutions, not merely equivalent representations in different coordinates. Combined with our decoder learning results—where optimal decoders for encoder codes show only $\sim 0.13$ cosine similarity to the original SAE decoder—this suggests neither the encoder's feature selection nor the trained dictionary directions are uniquely determined.

**Limitations** We study inference-time identifiability for fixed, pre-trained SAEs. We do not address training-time variance (different random seeds learning different features) or whether non-unique codes lead to different downstream interpretations. Our experiments use factual completion prompts; other domains may show different patterns. We also do not test whether alternative codes have different causal effects when used for activation steering.

**Future Work** Natural extensions include: (1) testing whether non-unique codes yield different feature interpretations, (2) causal intervention experiments comparing encoder vs alternative codes, and (3) developing methods to identify which features are robustly selected across algorithms. (4)Beyond SAEs, other interpretability methods face similar underdetermination problems, where an identifiability framework would provide an interesting characterization.

## 2.2 CONCLUSION

SAE representations are non-unique at multiple levels—different methods select different features, and multiple valid decoders exist—yet the encoder's codes are highly stable and linearly equivalent to other dictionary-based solutions. For mechanistic interpretability, this suggests caution when making claims about specific SAE features: they may be one valid decomposition among many, rather than the ground truth representation.

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

## A  SPARSE CODING ALGORITHMS

All methods operate on a fixed activation $\mathbf{x} \in \mathbb{R}^d$, frozen decoder $W_{\text{dec}} \in \mathbb{R}^{m \times d}$ with bias $\mathbf{b}_{\text{dec}}$, and target sparsity $k$ matching the encoder's $\|\mathbf{z}_{\text{enc}}\|_0$.

---

**Algorithm 1** Orthogonal Matching Pursuit (OMP)

---

**Require:** activation $\mathbf{x}$, decoder $W_{\text{dec}}$, bias $\mathbf{b}_{\text{dec}}$, sparsity $k$, excluded indices $E$ (optional)
**Ensure:** sparse code $\mathbf{z} \in \mathbb{R}^m$
1: $\mathbf{r} \leftarrow \mathbf{x} - \mathbf{b}_{\text{dec}}$                                                                 ▷ Initialize residual
2: $S \leftarrow \emptyset$                                                                                           ▷ Support set
3: $\mathbf{z} \leftarrow \mathbf{0} \in \mathbb{R}^m$
4: **for** $t = 1, \ldots, k$ **do**
5:     $c_j \leftarrow \langle W_{\text{dec},j}, \mathbf{r} \rangle \quad \forall j$                                    ▷ Correlations
6:     $c_j \leftarrow -\infty \quad \forall j \in S \cup E$                                                ▷ Mask selected/excluded
7:     $j^* \leftarrow \arg\max_j c_j$                                                                        ▷ Greedy selection
8:     $S \leftarrow S \cup \{j^*\}$
9:     $\mathbf{z}_S \leftarrow \text{LeastSquares}\left(W_{\text{dec},S}^\top, \mathbf{x} - \mathbf{b}_{\text{dec}}\right)$      ▷ Refit coefficients
10:     $\mathbf{r} \leftarrow \mathbf{x} - W_{\text{dec},S}^\top \mathbf{z}_S - \mathbf{b}_{\text{dec}}$               ▷ Update residual
11: **end for**
12: **return** $\mathbf{z}$ with support $S$

---

**Algorithm 2** Iterative Hard Thresholding (IHT)

---

**Require:** activation $\mathbf{x}$, decoder $W_{\text{dec}}$, bias $\mathbf{b}_{\text{dec}}$, sparsity $k$, step size $\eta$, iterations $T$, initialization mode
**Ensure:** sparse code $\mathbf{z} \in \mathbb{R}^m$
1: **if** init = encoder **then**
2:     $\mathbf{z} \leftarrow \mathbf{z}_{\text{enc}}$                                                                 ▷ Initialize from encoder
3: **else**
4:     $\mathbf{z} \leftarrow \mathbf{0}$
5:     sample $k$ indices uniformly; set $z_j \sim \text{Uniform}(0, 0.1)$ on sampled indices
6: **end if**
7: **for** $t = 1, \ldots, T$ **do**
8:     $\mathbf{g} \leftarrow -W_{\text{dec}}(\mathbf{x} - W_{\text{dec}}^\top \mathbf{z} - \mathbf{b}_{\text{dec}})$         ▷ Gradient
9:     $\mathbf{z} \leftarrow \mathbf{z} - \eta \mathbf{g}$                                                             ▷ Gradient step
10:     $\mathbf{z} \leftarrow \max(\mathbf{z}, 0)$                                                        ▷ Project to non-negative orthant
11:     $\mathbf{z} \leftarrow H_k(\mathbf{z})$                                                          ▷ Keep top-$k$ entries, zero rest
12: **end for**
13: **return** $\mathbf{z}$

---

We use $\eta = 0.05$ and $T = 300$. We have two initializations with one starting from the encoder's solution. This tests whether the encoder found a local optimum, and if so, IHT should stay near it and possibly refine the coefficients slightly. Our random initialization tests whether alternative local optima exist.

---

**Algorithm 3** Gradient Descent with Adaptive $\ell_1$ Penalty (GD-sparse)

---

**Require:** activation $\mathbf{x}$, decoder $W_{\mathrm{dec}}$, bias $\mathbf{b}_{\mathrm{dec}}$, target sparsity $k$, learning rate $\alpha$, iterations $T$
**Ensure:** sparse code $\mathbf{z} \in \mathbb{R}^m$
1: $\mathbf{z} \sim \mathrm{Uniform}(0, \sigma)^m$ where $\sigma = 0.01\|\mathbf{x}\|/\sqrt{m}$         ▷ Small random init
2: $\lambda \leftarrow 0.001$                                                 ▷ Initial $\ell_1$ penalty
3: **for** $t = 1, \ldots, T$ **do**
4:     $\mathcal{L} \leftarrow \|\mathbf{x} - W_{\mathrm{dec}}^\top \mathbf{z} - \mathbf{b}_{\mathrm{dec}}\|^2 + \lambda\|\mathbf{z}\|_1$
5:     $\mathbf{z} \leftarrow \mathrm{AdamStep}(\mathbf{z}, \nabla_{\mathbf{z}}\mathcal{L}, \alpha)$         ▷ $\alpha = 10^{-3}$, gradient clipped to norm 1
6:     $\mathbf{z} \leftarrow \max(\mathbf{z}, 0)$                              ▷ Project to non-negative orthant
7:     **if** $t \bmod 50 = 0$ **then**                                        ▷ Adaptive $\lambda$ schedule
8:         **if** $\|\mathbf{z}\|_0 > 1.2k$ **then**
9:             $\lambda \leftarrow 1.5\lambda$
10:         **else if** $\|\mathbf{z}\|_0 < 0.8k$ **then**
11:             $\lambda \leftarrow 0.7\lambda$
12:         **end if**
13:     **end if**
14: **end for**
15: $\mathbf{z} \leftarrow H_k(\mathbf{z})$                                    ▷ Final hard threshold to exactly $k$
16: **return** $\mathbf{z}$

---

We use $T = 500$. Optimization is performed in float32 for numerical stability. Note that the final hard-thresholding step (line 12) can substantially degrade reconstruction quality, as features optimized jointly under soft sparsity may not remain optimal when pruned.

## B    RECONSTRUCTION LOSS AND LINEAR IDENTIFIABILITY METRICS

Table 2: Sparse coding results on GemmaScope SAEs (Gemma 2 2B). All metrics compare encoder vs. each method.

| SAE | Layer | OMP | | | | | IHT | | | | | GD | | | | |
|-----|-------|------|-------|------|------|------|------|-------|------|------|------|------|-------|------|------|------|
| | | Jacc | $R^2$ | Cos | CCA | Proc | Jacc | $R^2$ | Cos | CCA | Proc | Jacc | $R^2$ | Cos | CCA | Proc |
| 2B-16k | L5 | 0.48 | 0.90 | 0.91 | 1.00 | 0.22 | 0.45 | 0.93 | 0.93 | 1.00 | 0.18 | 0.01 | 0.52 | 0.03 | 0.99 | 16.7 |
| 2B-16k | L12 | 0.41 | 0.88 | 0.86 | 1.00 | 0.24 | 0.43 | 0.90 | 0.89 | 1.00 | 0.21 | 0.00 | 0.48 | 0.00 | 0.98 | 18.4 |
| 2B-16k | L20 | 0.43 | 0.91 | 0.87 | 1.00 | 0.23 | 0.48 | 0.94 | 0.91 | 1.00 | 0.17 | 0.00 | 0.26 | 0.00 | 0.84 | 47.9 |
| 2B-65k | L20 | 0.38 | 0.90 | 0.85 | 1.00 | 0.25 | 0.42 | 0.91 | 0.89 | 1.00 | 0.24 | 0.00 | 0.31 | 0.01 | 0.90 | 58.5 |

Table 3: OMP with forced feature exclusion (Gemma 2 2B, 16k width). Metrics compare encoder vs OMP variants.

| OMP | L5 | | | | | L12 | | | | |
|-----|------|-------|------|------|------|------|-------|------|------|------|
| | Jacc | $R^2$ | Cos | CCA | Proc | Jacc | $R^2$ | Cos | CCA | Proc |
| Free | 0.48 | 0.90 | 0.91 | 1.00 | 0.22 | 0.41 | 0.88 | 0.86 | 1.00 | 0.24 |
| $-$Top1 | 0.41 | 0.84 | 0.67 | 1.00 | 0.39 | 0.38 | 0.79 | 0.72 | 1.00 | 0.42 |
| $-$Top2 | 0.39 | 0.81 | 0.56 | 1.00 | 0.43 | 0.36 | 0.77 | 0.63 | 1.00 | 0.46 |
| $-$Top3 | 0.37 | 0.80 | 0.50 | 1.00 | 0.47 | 0.35 | 0.77 | 0.57 | 1.00 | 0.44 |

| OMP | L20 (16k) | | | | | L20 (65k) | | | | |
|-----|------|-------|------|------|------|------|-------|------|------|------|
| | Jacc | $R^2$ | Cos | CCA | Proc | Jacc | $R^2$ | Cos | CCA | Proc |
| Free | 0.43 | 0.91 | 0.87 | 1.00 | 0.23 | 0.38 | 0.90 | 0.85 | 1.00 | 0.25 |
| $-$Top1 | 0.41 | 0.80 | 0.74 | 1.00 | 0.42 | 0.35 | 0.80 | 0.70 | 1.00 | 0.45 |
| $-$Top2 | 0.39 | 0.77 | 0.66 | 0.99 | 0.47 | 0.33 | 0.77 | 0.62 | 1.00 | 0.50 |
| $-$Top3 | 0.38 | 0.76 | 0.61 | 0.99 | 0.49 | 0.32 | 0.75 | 0.56 | 0.99 | 0.52 |

For OMP R² stays high (0.75–0.80) even when excluding top features, while Cosine drops sharply (0.91→0.50). This appears to indicate codes remain linearly related but geometrically different.

## C    RECONSTRUCTION QUALITY

A notable finding is that OMP consistently achieves better reconstruction than the encoder despite selecting largely different features. Table 2 shows OMP achieving relative reconstruction loss of 0.93 across layers—a 7% improvement over the encoder baseline—while sharing only $\sim 43\%$ of features (Jaccard similarity).

This indicates the encoder's solution is not merely "one valid decomposition among many," but is actually suboptimal for reconstruction. The encoder's one-shot amortized inference trades reconstruction quality for computational efficiency. OMP's sequential greedy selection, while slower, finds genuinely better solutions in the same dictionary.

IHT achieves reconstruction quality comparable to the encoder (relative loss $\sim 0.99$–$1.02$), suggesting the encoder's solution lies near a local optimum that IHT can refine slightly. When initialized randomly, IHT converges to different features (Jaccard $\sim 0.45$) with similar reconstruction quality, confirming multiple valid optima exist.

### GD-SPARSE INSTABILITY

Unlike OMP and IHT, gradient descent with soft $\ell_1$ constraints proved unstable in our setting, converging to qualitatively different solutions (Table 2: $R^2 = 0.26$–$0.52$, Jaccard $\approx 0$). We attribute this to the mismatch between soft and hard sparsity constraints: features optimized jointly under $\ell_1$ penalties may not remain optimal when subsequently thresholded to exactly $k$ active features. The hard thresholding step—necessary for fair comparison—discards features that the optimization relied upon, degrading reconstruction by approximately $2\times$.

This instability is itself informative: it suggests the sparse coding landscape contains multiple qualitatively distinct basins, not merely coordinate ambiguity within a shared subspace. Methods that maintain hard sparsity throughout (OMP, IHT) find solutions linearly equivalent to the encoder; methods that relax this constraint during optimization may escape to different basins entirely.

## D    LINEAR IDENTIFIABILITY METRICS

### BACKGROUND: LINEAR IDENTIFIABILITY

We adopt the framework of Roeder et al. (2021), who prove that learned representations are *linearly identifiable*: models trained on the same data learn representations that are equal up to an invertible linear transformation. Formally, if $f_1$ and $f_2$ are representation functions learned by different training runs, there exists an invertible matrix $A$ such that:

$$f_1(x) = A \cdot f_2(x)$$

for all inputs $x$.

For representation learning generally, linear identifiability could be regarded as a positive result— different training runs converge to the same "information content," merely expressed in different coordinate systems. However, for mechanistic interpretability, linear identifiability may be insufficient. When we claim "feature 7493 represents dogs," we are making a claim about a specific dictionary element, not merely the subspace it contributes to. Two codes can be perfectly linearly equivalent while using entirely different features.

We therefore distinguish two forms of identifiability:

- **Linear identifiability:** Codes contain equivalent information, recoverable via linear transformation.
- **Support identifiability:** Codes select the same dictionary elements (same sparsity pattern).

Our experiments reveal these can dissociate: linear identifiability holds while support identifiability fails.

METRICS

**Canonical Correlation Analysis (CCA).** We use CCA to measure linear similarity between code sets. Given codes $Z_1 \in \mathbb{R}^{n \times m}$ from method 1 and $Z_2 \in \mathbb{R}^{n \times m}$ from method 2, CCA finds optimal linear projections $C$ and $D$ such that the pairwise correlations $\rho_i = \text{Corr}(C_i^\top Z_1, D_i^\top Z_2)$ are maximized. If one code set is a linear transform of another, CCA will recover this transformation and the mean of $\rho$ will approach 1. We report the mean CCA coefficient; values $\approx 1$ indicate codes live in the same linear subspace.

**Bidirectional $R^2$.** We fit linear regressions in both directions—predicting $z_{\text{enc}}$ from $z_{\text{alt}}$ and vice versa—and average the $R^2$ values. High bidirectional $R^2$ indicates mutual linear predictability. Unlike CCA, which finds optimal projections, $R^2$ directly measures how much variance one code explains in the other. Note that $R^2 \approx 1$ is possible even when codes use completely different features, as long as those features span equivalent subspaces.

**Cosine Similarity.** For each activation, we compute:

$$\cos(z_{\text{enc}}, z_{\text{alt}}) = \frac{z_{\text{enc}} \cdot z_{\text{alt}}}{\|z_{\text{enc}}\| \|z_{\text{alt}}\|}$$

and average across activations. High cosine similarity indicates codes point in similar geometric directions. Interestingly, high cosine with low Jaccard suggests dictionary redundancy: different feature combinations achieving similar directions.

**Jaccard Similarity (Support Overlap).** Given supports $S_{\text{enc}} = \{i : z_{\text{enc},i} > 0\}$ and $S_{\text{alt}}$, Jaccard similarity is:

$$J = \frac{|S_{\text{enc}} \cap S_{\text{alt}}|}{|S_{\text{enc}} \cup S_{\text{alt}}|}$$

This directly measures whether methods select the *same* dictionary elements, independent of coefficient values. $J = 1$ means identical feature selection; $J = 0$ means completely disjoint supports.

**Shared Support Correlation.** For features selected by *both* methods ($i \in S_{\text{enc}} \cap S_{\text{alt}}$), we compute the Pearson correlation between their coefficients.

### D.1 HANDLING THE $n \ll m$ PROBLEM

With $n \approx 600$ samples and $m = 16,384$ features, direct linear regression is massively underdetermined. Ordinary least squares will find a perfect fit ($R^2 = 1$) even for completely unrelated data, yielding spurious results.

Roeder et al. (2021) note a related issue: deep neural network representations concentrate most variability in a low-dimensional subspace, leaving many noisy dimensions that can cause spurious high correlations in CCA. Their solution, SVCCA (**?**), applies PCA before CCA.

We adopt a similar approach: before computing $R^2$ or CCA, we project both code matrices onto their top 50 principal components, retaining approximately $85\%$ of variance. This serves two purposes:

1. **Regularization:** By restricting to the principal subspace, we ensure metrics reflect genuine structure in the codes' active manifold rather than overfitting artifacts.

2. **Computational tractability:** CCA and linear regression in 50 dimensions are well-conditioned, whereas 16,384-dimensional regression with 500 samples is not.

### D.2 INTERPRETING THE DISSOCIATION

Our central finding is that linear metrics (CCA $> 0.99$, $R^2 > 0.88$) indicate equivalence while support metrics (Jaccard $\sim 0.43$) indicate disagreement. The dissociation implies **dictionary redundancy**: multiple dictionary elements can substitute for each other in reconstruction. When encoder and OMP both achieve high-quality reconstruction using largely different features, those feature sets must span similar subspaces. The dictionary contains "near-synonyms"—features that are mathematically interchangeable for reconstruction purposes.

This is consistent with the overcomplete nature of SAE dictionaries ($m \gg d$, typically 8–32×). The encoder and OMP simply resolve this redundancy differently. For interpretability, this raises a question: if feature 7493 and feature 2841 are "synonyms," which one truly "represents" the underlying concept? Reconstruction loss cannot distinguish them.

# E  Noise Stability Experiments

## E.1  Motivation

The existence of alternative sparse codes raises a natural question: is the encoder's solution stable, or can small perturbations cause large changes? If the encoder's codes are highly sensitive to noise, the observed non-uniqueness across methods might simply reflect optimization instability rather than genuine multiplicity of solutions.

We test this by adding Gaussian noise to activations and measuring how sparse codes change. Formally, given activation $x$, we compute $x_{\text{noisy}} = x + \sigma\epsilon$ where $\epsilon \sim \mathcal{N}(0, I)$, then measure Jaccard similarity between $z(x)$ and $z(x_{\text{noisy}})$.

## E.2  Experimental Setup

We test noise levels $\sigma \in \{0.01, 0.02, 0.05, 0.1, 0.2, 0.5, 1.0\}$ across layers 5 and 12 of Gemma 2 2B with 16k-width GemmaScope SAEs. For reference, mean activation norms are $91.1$ (layer 5) and $144.9$ (layer 12), so $\sigma = 0.1$ corresponds to SNR $\approx 900$–$1400$ (59–63 dB)—a relatively small perturbation.

For each method (encoder, OMP, IHT, GD-sparse), we:

1. Compute clean codes $z(x)$ for all activations.
2. Add noise and compute noisy codes $z(x_{\text{noisy}})$.
3. Measure Jaccard similarity between clean and noisy supports.

## E.3  Results

Table 4 shows Jaccard similarity between clean and noisy codes across noise levels.

Table 4: Code stability under Gaussian noise (Jaccard with clean codes).

| $\sigma$ | **Layer 5** (Norm: 91.1) | | | | **Layer 12** (Norm: 144.9) | | | |
| --- | --- | --- | --- | --- | --- | --- | --- | --- |
| | Encoder | OMP | IHT | GD | Encoder | OMP | IHT | GD |
| 0.01 | 0.994 | 0.958 | 0.935 | 0.837 | 0.996 | 0.924 | 0.933 | 0.901 |
| 0.02 | 0.988 | 0.929 | 0.929 | 0.779 | 0.991 | 0.891 | 0.930 | 0.863 |
| 0.05 | 0.971 | 0.885 | 0.909 | 0.677 | 0.978 | 0.845 | 0.917 | 0.788 |
| 0.10 | 0.940 | 0.839 | 0.878 | 0.565 | 0.955 | 0.808 | 0.899 | 0.705 |
| 0.20 | 0.880 | 0.766 | 0.818 | 0.432 | 0.913 | 0.758 | 0.857 | 0.594 |
| 0.50 | 0.695 | 0.580 | 0.666 | 0.241 | 0.788 | 0.639 | 0.746 | 0.402 |
| 1.00 | 0.366 | 0.339 | 0.489 | 0.125 | 0.587 | 0.459 | 0.602 | 0.246 |

## E.4  Key Findings

- **The encoder is highly stable.** At $\sigma = 0.1$, encoder codes maintain Jaccard $> 0.94$ with their clean counterparts. The encoder consistently finds its particular solution; the observed non-uniqueness across methods is not due to the encoder being unstable.

- **Stability ordering: Encoder $>$ IHT $>$ OMP $>$ GD.** The encoder's amortized inference produces the most stable codes. IHT, which maintains hard sparsity constraints throughout optimization, is second most stable. OMP's greedy sequential selection is more sensitive to small changes in correlations.

- **Non-uniqueness reflects multiple valid solutions, not instability.** This is the central finding. Despite the encoder achieving Jaccard $> 0.94$ under noise (i.e., highly stable), it still shows only Jaccard $\sim 0.43$ agreement with OMP on clean activations. The encoder reliably finds *its* solution; OMP reliably finds a *different* solution. Both are stable, but they converge to different basins.

## E.5  Self-Consistency Summary

At $\sigma = 0.1$, self-consistency scores (Jaccard with own clean codes):

| Layer | Encoder | OMP | IHT | GD |
|-------|---------|-----|-----|-----|
| 5 | 0.940 | 0.839 | 0.878 | 0.565 |
| 12 | 0.955 | 0.808 | 0.899 | 0.705 |

The encoder's high self-consistency combined with low cross-method agreement allows us to see that the sparse coding landscape contains multiple stable, distinct solutions.

## F   DECODER NON-UNIQUENESS

### F.1   MOTIVATION

Our main experiments evaluate whether sparse codes are uniquely determined for a fixed decoder. But a deeper question is whether the *decoder itself* is uniquely determined. Given activations $X$ and codes $Z$ from any method, we can ask: what decoder $D^*$ optimally reconstructs $X$ from $Z$? If $D^*$ closely matches the original SAE decoder $W_{\text{dec}}$, the representation is well-identified. If $D^*$ differs substantially, non-uniqueness extends to the dictionary directions themselves.

### F.2   METHOD

For each sparse coding method and sparsity level $k$, we:

1. Obtain codes $Z \in \mathbb{R}^{n \times m}$ for $n$ activations.
2. Learn the optimal decoder via ridge regression:
$$D^* = \arg\min_D \|X - ZD\|^2 + \lambda\|D\|^2$$
   where $\lambda = 10^{-6}$ provides numerical stability.
3. Compare learned decoder rows to original SAE decoder rows via cosine similarity.

We test $k \in \{2, 4, 8, 16, 32, 64\}$ across layers 5, 12, and 20 of Gemma 2 2B with 16k-width SAEs.

### F.3   RESULTS

Table 5 shows cosine similarity between learned and original decoder rows, averaged over active features. Table 6 compares optimal decoders between different methods.

Table 5: Learned decoder similarity to original SAE decoder (Cosine Similarity).

| | **Layer 5** | | | | **Layer 12** | | | | **Layer 20** | | | |
|----|-----|-----|-----|-----|-----|-----|-----|-----|-----|-----|-----|-----|
| $k$ | Enc | OMP | IHT | GD | Enc | OMP | IHT | GD | Enc | OMP | IHT | GD |
| 2 | 0.25 | 0.26 | 0.25 | 0.07 | 0.32 | 0.21 | 0.22 | 0.01 | 0.27 | 0.27 | 0.25 | 0.00 |
| 4 | 0.13 | 0.12 | 0.13 | 0.01 | 0.16 | 0.14 | 0.11 | 0.02 | 0.12 | 0.10 | 0.09 | 0.01 |
| 8 | 0.07 | 0.07 | 0.07 | 0.01 | 0.08 | 0.06 | 0.06 | 0.00 | 0.10 | 0.06 | 0.02 | 0.01 |
| 16 | 0.04 | 0.01 | 0.03 | 0.01 | 0.02 | 0.01 | 0.02 | 0.01 | 0.22 | 0.24 | 0.22 | 0.01 |
| 32 | 0.15 | 0.18 | 0.11 | 0.02 | 0.11 | 0.15 | 0.11 | 0.02 | 0.22 | 0.21 | 0.21 | 0.01 |
| 64 | 0.16 | 0.12 | 0.16 | 0.02 | 0.15 | 0.12 | 0.14 | 0.02 | 0.18 | 0.14 | 0.19 | 0.02 |

Table 6: Cross-method decoder similarity (cosine similarity for shared features).

| Layer | Enc-OMP | Enc-IHT | Enc-GD | OMP-IHT | OMP-GD | IHT-GD |
|-------|---------|---------|--------|---------|--------|--------|
| 5 | 0.34 | 0.30 | 0.07 | 0.34 | 0.11 | 0.09 |
| 12 | 0.34 | 0.33 | 0.15 | 0.36 | 0.11 | 0.10 |
| 20 | 0.48 | 0.45 | 0.09 | 0.48 | 0.09 | 0.05 |

KEY FINDINGS

- **Learned decoders differ substantially from original.** Across all methods and layers, cosine similarity between learned and original decoders averages only 0.10–0.20. The

optimal decoder for reconstructing activations from sparse codes is *not* the SAE's trained decoder. Even when using the encoder's own codes, the learned optimal directions differ from $W_{dec}$.

- **Cross-method agreement is moderate.** When encoder and OMP both select a feature, their learned optimal directions for that feature show $\sim 0.34$–$0.48$ cosine similarity. This suggests different codes lead to different optimal decoders even for shared features.

- **GD-sparse produces decoders near-orthogonal to original.** GD-sparse codes yield learned decoders with near-zero similarity to the original ($\sim 0.01$–$0.02$) and low cross-method agreement ($\sim 0.05$–$0.15$).

## G  CAUSAL ABLATION EXPERIMENTS

### G.1  MOTIVATION

Experiments in the main text establish that alternative sparse codes exist and are linearly equivalent but select different features. A natural question is whether this matters for model behavior. If different codes have different causal effects when used for interventions, the choice of sparse coding method has downstream consequences for interpretability analyses.

### G.2  INTERVENTION PROTOCOL

For each prompt where Gemma 2 2B answered correctly, we:

1. Extract the layer-20 residual stream activation $x$ at the final token position.

2. Compute sparse codes $z$ using each method (encoder, OMP, IHT variants).

3. Ablate the top-$k$ features (by coefficient magnitude) by subtracting their contributions:

$$x_{ablated} = x - \sum_{i \in \text{top-}k} z_i \cdot d_i$$

4. Run the model with the modified activation and measure behavioral change.

We report two metrics:

- **Breaking rate:** Fraction of prompts where ablation flips the model's top-1 prediction from correct to incorrect.

- **Probability drop:** Decrease in probability assigned to the correct answer, $P_{baseline}(\text{target}) - P_{ablated}(\text{target})$.

### G.3  RESULTS

Table 7 shows causal ablation results at $k = 5$ on the full dataset ($\sim 600$ prompts). Table 8 shows results across varying $k$ levels for a smaller subset.

Table 7: Causal ablation results ($k = 5$ features ablated, layer 20).

| Method | Breaking Rate | Prob Drop | Jaccard | Rel. Recon Loss |
|---|---|---|---|---|
| Encoder | 40.5% | 0.178 | 1.000 | 1.000 |
| OMP | 42.9% | 0.195 | 0.450 | 0.928 |
| IHT (from encoder) | 41.1% | 0.188 | 0.992 | 0.976 |
| IHT (random init) | 41.1% | 0.175 | 0.489 | 0.988 |
| OMP-exclude-top1 | 32.9% | 0.137 | 0.423 | 1.060 |
| OMP-exclude-top2 | 32.3% | 0.131 | 0.405 | 1.113 |
| OMP-exclude-top3 | 26.6% | 0.104 | 0.391 | 1.148 |

Table 8: Breaking rate by sparsity level ($n = 25$, preliminary).

| Method | $k = 1$ | $k = 3$ | $k = 5$ | $k = 10$ | $k = 20$ |
|---|---|---|---|---|---|
| Encoder | 3.8% | 26.9% | 34.6% | 38.5% | 34.6% |
| OMP | 11.5% | 30.8% | 46.2% | 50.0% | 57.7% |
| IHT (from encoder) | 3.8% | 30.8% | 50.0% | 34.6% | 42.3% |
| IHT (random init) | 3.8% | 23.1% | 42.3% | 30.8% | 34.6% |

### G.4 KEY FINDINGS

- **Different features, comparable or greater causal impact.** OMP achieves 7% better reconstruction while using a largely disjoint feature set ($J = 0.45$), yet its top features break the model more often than the encoder's (42.9% vs 40.5%).

- **Alternative optima have comparable aggregate impact.** Two decompositions can have equivalent aggregate causal importance while disagreeing on which specific features matter. Different valid decompositions could support different mechanistic narratives.

- **The encoder's features aren't uniquely necessary.** Even when OMP is forbidden from using the encoder's top features, breaking rates remain substantial (27–33%).

### LIMITATIONS

Ablation as an intervention is imperfect, as it may conflate feature importance with general computational disruption. Our analysis is focused on a single layer ($L20$) and factual completion tasks; broader coverage across domains and the addition of activation steering experiments would further strengthen these conclusions.

