# OpenReview forum: "Which Sparse Code? Identifiability Failures in SAE Inference"
_ICLR.cc/2026/Workshop/Sci4DL — Sci4DL 2026_

### Official Review · Reviewer_u6Rb · 2026-02-25

**Fit:** 3
**Significance:** 2
**Confidence:** 3

**Summary:**

Sparse autoencoders (SAEs) are widely used in mechanistic interpretability to visualize the latent space of LLM. SAE is used to extract interpretable sparse feature from dense LLM embedding, The author shows the sparse feature provided by SAE is not identifiable. Other sparse coding algorithms (OMP, IHT) provide different sparse feature.

**Strengths:**

The author identify an important issue for the popular tool in mechanistic interpretability (area for understanding deep learning model). Interpretable feature from SAE might be un-identifiable/un-stable.

**Suggestions:**

The mathematically statement in this paper is a bit loose. What do the author mean by unique? The idea behind toy model of superposition is that the ground-truth representation of concept space is sparse and high dimensional, but LLM compress it into a lower dimension via linear transform. Under this assumption, the SAE tries to solve the L0-sparse coding, find sparse vector that when compress, it reconstructs the LLM embedding vector [\min_z ||x - Dz||^2 + ||z||_0]. Under this objective, the question you want to ask is that is the solution of the L0-sparse coding unique? The answer is no because the L0 sparse coding problem is not convex, and there's many approximated solver like: SAE, IHT, OMP...

One way to make it identifiable is to relax the L0 sparse coding problem to the L1 sparse coding problem, then this become a convex optimization problem with many solver that provide identifiable solution like ISTA, FISTA. This is used in the original LLM feature visualization paper, which is not cited by the author [1].

Similarly, the question in the second page is also misleading, the author ask "Does non-uniqueness extend to decoders?" And fitting a linear regression between the sparse feature and LLM activation, and call this the "decoder." This is not the decoder. The decoder is the decoder learned in SAE. And the SAE decoder is well-known to be un-identifiable because sparse dictionary learning is un-identifiable.

[1] Yun et al. 2021. Transformer visualization via dictionary learning: contextualized embedding as a linear superposition of transformer factors

---

### Meta-Review · Area_Chair_Z7D4 · 2026-03-01

**Recommendation:** Accept

**Metareview:**

Misses a lot of related work, e.g.

1. Neel Nanda et al. — *Progress Update #1 from the GDM Mech Interp Team* (Alignment Forum, 2024)
2. O’Neill, Gumran, Klindt — *Compute Optimal Inference and Provable Amortisation Gap in Sparse Autoencoders* (arXiv:2411.13117, 2024)
3. Leask, Nanda, Al Moubayed — *Inference-Time Decomposition of Activations (ITDA): A Scalable Approach to Interpreting Large Language Models* (arXiv:2505.17769, 2025)
4. Rajamanoharan et al. — *Improving Dictionary Learning with Gated Sparse Autoencoders* (2024)
5. Chanin et al. — *A is for Absorption: Studying Feature Splitting and Absorption in Sparse Autoencoders* (arXiv:2409.14507, 2024)
6. Engels, Riggs, Tegmark — *Decomposing The Dark Matter of Sparse Autoencoders* (arXiv:2410.14670, 2024)

---

### Decision · Program_Chairs · 2026-03-02

Accept